# Coupled Tensor Block Term Decomposition with Superpixel-Based Graph Laplacian Regularization for Hyperspectral Super-Resolution

**Hongyi Liu** [1,*]🆔, **Wen Jiang** [1], **Yuchen Zha** [1] **and Zhihui Wei** [2]

[1] School of Mathematics and Statistics, Nanjing University of Science and Technology, Nanjing 210094, China

[2] School of Computer Science and Engineering, Nanjing University of Science and Technology, Nanjing 210094, China

[*] Correspondence: hyliu@njust.edu.cn

**Abstract:** Hyperspectral image (HSI) super-resolution aims at improving the spatial resolution of HSI by fusing a high spatial resolution multispectral image (MSI). To preserve local submanifold structures in HSI super-resolution, a novel superpixel graph-based super-resolution method is proposed. Firstly, the MSI is segmented into superpixel blocks to form two-directional feature tensors, then two graphs are created using spectral–spatial distance between the unfolded feature tensors. Secondly, two graph Laplacian terms involving underlying BTD factors of high-resolution HSI are developed, which ensures the inheritance of the spatial geometric structures. Finally, by incorporating graph Laplacian priors with the coupled BTD degradation model, a HSI super-resolution model is established. Experimental results demonstrate that the proposed method achieves better fused results compared with other advanced super-resolution methods, especially on the improvement of the spatial structure.

**Keywords:** hyperspectral image; multispectral image; super-resolution; tensor block term decomposition; graph Laplacian regularization; superpixel segmentation

## 1. Introduction

Hyperspectral remote sensing has been widely used in the fields of geological exploration, agricultural production, urban planning, environmental monitoring and so on. Due to the limitation of the optical imaging mechanism of airborne spectrometers, hyperspectral image (HSI) with high spectral resolution is often accompanied by lower spatial resolution, which brings inconvenience to the applications, such as classification, anomaly detection, and object recognition [1,2]. Therefore, improving the spatial resolution of HSI has become an urgent issue. Fortunately, multispectral image (MSI) has high spatial resolution but with low spectral resolution. Therefore, the fusion of HSI and MSI (HSI–MSI), also called HSI super-resolution, provides an effective and efficient way to improve the spatial resolution of HSI and results in a high-resolution hyperspectral image (HR-HSI). Furthermore, by fusing HSI–MSI, the shortcoming of a single imaging device in spatial-spectral resolution has been overcome, and a more comprehensive and accurate understanding of the observed environment can be obtained [3,4].

### 1.1. Relates Works

In general, the HSI–MSI fusion methods can be categorized as four classes [5–7], pansharpening-based methods [8,9], matrix-based methods [10–12], tensor-based methods [13–17], and deep CNN-based methods [18,19]. Among them, matrix and tensor methods depend on the degradation model of high-resolution HSI combined with priors of the decomposition matrix or tensor factors. Regularization priors, such as low-rank [20–24], sparsity [25–27], or graph Laplacian [28,29], are widely used in HSI–MSI fusion models.

However, in matrix-based methods, the HSI is unfolded into matrices, which ignores the inherent three-dimensional features. Tensor, as a multi-dimension array, provides a flexible representation of HSI and has made tensor-based HSI super-resolution methods popular recently.

In tensor-based HSI super-resolution, the HSI is modeled as a third-order tensor with two spatial dimensions and one spectral dimension, which can fully exploit the dependence across different dimensions or modes. Moreover, tensor has more flexible decomposition forms, such as canonical polyadic decomposition (CPD) [30], Tucker decomposition (TD) [23], singular value decomposition (t-SVD) [31,32], tensor ring decomposition (TRD) [33,34], tensor block term decomposition (BTD) [35,36], etc. Each decomposition leads to a different perspective in understanding the correlation between the different modes of the tensor. Canonical polyadic decomposition (CPD) represents a tensor with a sum of $R$ tensors, in which each tensor is rank-1. Kanatsoulis et al. [37] first formulated a coupled tensor CPD-based HSI–MSI fusion model by establishing the relations between tensor mode-product and the HSI degradation model. Furthermore, the blind and semi-blind HSI–MSI fusion models were solved by the alternative optimization algorithm, which is a well-known STEREO algorithm. Xu et al. [38] proposed a HSI super-resolution model based on non-local coupled tensor patches, in which the constructed fourth-order low-rank tensor is guided by MSI, and the HR-HSI and MSI share the same nonlocal tensor CPD factor matrices. Although CPD can represent the three-dimensional data structure, it has a high computational complexity and the CP rank $R$ is hard to compute. Tucker decomposition (TD) provides a flexible decomposition with a core tensor and three matrix factors. Each factor of TD is easy to compute. Assuming the HR-HSI has a low Tucker multilinear rank, a coupled TD-based HSI–MSI fusion model with blind and semi-blind SCOTT algorithms is proposed in [16]. Furthermore, the decomposed core tensor was estimated by solving the generalized Sylvester equation, and three factor matrices were computed by truncated singular value decomposition. In addition, TD factors can be viewed from spatial and spectral dimensions, so a dictionary learning strategy was introduced to train spatial and spectral dictionaries from MSI and low-resolution HSI, respectively [13]. Furthermore, by considering the spectral smoothness and spatial consistency as priors, a graph regularized low-rank tensor fusion method was developed in [17]. Borsoi et al. [39] assumed that there is spectral variability in HSI–MSI fusion and introduced the variability into the TD fusion model with an additive term. In general, tensor CPD and TD can be unified as tensor BTD. Specifically, tensor BTD provides a clear physical explanation for the factor matrices from the perspective of unmixing, and makes the prior of the abundance and endmember easy to model [40–42]. Therefore, a coupled BTD-based HSI–MSI fusion technique has become popular for the linear unmixing model involved [43,44]. The endmember and low-rank abundance map are represented by potential factor matrices under BTD. However, due to the lack of constraints on the factor matrix, the quality of the fused HSI is reduced. Therefore, other regularization terms have the potential to be combined with BTD in fusion models to improve the recovery performance.

### 1.2. Motivations and Contributions

In terms of regularization, factors such as total variation [45], low-rank, sparse, graph Laplacian [46], etc. have been widely exploited combined with tensor decomposition as mentioned. Among them, manifold Laplacian is an effective strategy to improve the spatial structure of images. In [46], a graph Laplacian is integrated with BTD for HSI–MSI fusion, resulting in an impressive performance. However, the local weight matrix of the graph is calculated in a pixel-wise manner, which can easily be affected by noise and cannot describe the local geometry of the image well. Therefore, to exploit the spatial neighborhood structure by generating homogeneous segmented regions and reduce the sensitivity to noise and outliers, a superpixel-guided graph Laplacian regularization is constructed in this paper. Furthermore, considering the merits of the BTD, the graph Laplacian is introduced to the BTD-based HSI–MSI fusion framework, resulting in the

regularization of the proposed superpixel-based graph Laplacian with the BTD fusion method, named as SGLCBTD for short. The main contributions are as follows.

(1) The MSI is segmented by regional clustering according to spectral–spatial distance measurements;
(2) Two-directional tensor graphs are designed via the features of the segmented MSI superpixel blocks, whose local geometric structure is consistent with HSI;
(3) The similarity weights of the superpixel blocks are calculated and graph Laplacian matrices are constructed, which is used to convey the spatial manifold structures from MSI to the factor matrices of HSI;
(4) The proposed superpixel graph Laplacian BTD model is solved by the block coordinate descent algorithm, and the experimental results are displayed.

## 2. Background

For convenience, some necessary definitions and preliminaries of tensor are introduced first. A scalar, a vector, a matrix, and a tensor are denoted as $x, \boldsymbol{x}, \boldsymbol{X}$ and $\boldsymbol{\mathcal{X}}$, respectively. $x_i, \boldsymbol{X}_{ij}$ and $\boldsymbol{\mathcal{X}}_{ijk}$ denote the $i$-th, $(i,j)$-th and $(i,j,k)$-th element of $\boldsymbol{x} \in \mathbb{R}^I, X \in \mathbb{R}^{I \times J}$ and $\boldsymbol{\mathcal{X}} \in \mathbb{R}^{I \times J \times K}$, respectively. The n-mode unfolding of tensor $\boldsymbol{\mathcal{X}}$ is represented by $\boldsymbol{X}^{(n)}$. The one-mode product of tensor $\boldsymbol{\mathcal{X}} \in \mathbb{R}^{I \times J \times K}$ with a matrix $A \in \mathbb{R}^{M \times I}$ is denoted by $\boldsymbol{\mathcal{Y}} = \boldsymbol{\mathcal{X}} \times_1 A = AX^{(1)}$, thus, for the $n$-mode products, $n = 1, 2, 3$. $A \circ \boldsymbol{x}$ stands for the outer product of a matrix $A \in \mathbb{R}^{M \times N}$ and a vector $\boldsymbol{x} \in \mathbb{R}^I$, resulting in an $M \times N \times I$ tensor. For two matrices $A \in \mathbb{R}^{I \times K}$ and $B \in \mathbb{R}^{I \times K}$, the Khatri–Rao product is $A \odot B \in \mathbb{R}^{IJ \times K}$, and $\odot_p$ stands for the column-wise Khatri–Rao product along the column.

### 2.1. Block Term Decomposition

The block term decomposition (BTD) in rank-$(L_r, L_r, 1)$. terms is defined that a third-order tensor $\boldsymbol{\mathcal{X}} \in \mathbb{R}^{I \times J \times K}$. can be decomposed as the sum of rank-$(L_r, L_r, 1)$ terms [35]

$$\boldsymbol{\mathcal{X}} \approx \sum_{r=1}^{R} \left( A_r B_r^T \right) \circ c_r \tag{1}$$

where $A_r \in \mathbb{R}^{I \times L_r}$  $B_r \in \mathbb{R}^{J \times L_r}$ are full-column matrices with rank-$Lr$, and $c_r \in \mathbb{R}^K$, $R$ is the rank of the tensor $\boldsymbol{\mathcal{X}}$.

Let $= B_r^T \in \mathbb{R}^{I \times J}$ and replace it into formula (1), so

$$\boldsymbol{\mathcal{X}} \approx \sum_{r=1}^{R} S_r \circ c_r \tag{2}$$

Furthermore, unfolding the tensor along the third mode results in the following expression from the matrix point of view:

$$X^{(3)} = CS + \varepsilon \tag{3}$$

where $X^{(3)} \in \mathbb{R}^{K \times IJ}$ is the three-mode unfolding of tensor $\boldsymbol{\mathcal{X}}$, matrices $C = [c_1, \cdots, c_R] \in \mathbb{R}^{K \times R}$ and $S = [S(1,:), \cdots, S(R,:)] \in \mathbb{R}^{R \times IJ}$ have rank-$R$, each element $S(r,:) \in \mathbb{R}^{IJ}$ of $S$ constitutes the matrix $S_r$ with size of $I \times J$, the error term $\varepsilon$ is the Gaussian noise.

The Formula (3) can be easily connected with the linear unmixing model (LMM) of HSI. Specifically, matrix $C$ is the endmember matrix containing the spectral signatures of $R$ endmembers $c_1, c_2, \ldots c_R$, matrix $S$ represents the abundance coefficient matrix, each $S_r$ is the abundance corresponding to endmember $c_r$. Therefore, decomposition factors $A_r$ and $B_r$ of matrix $S_r$ also represent the spatial abundance information.

### 2.2. Problem Formulation

Let $\boldsymbol{\mathcal{Y}}_h \in \mathbb{R}^{I_h \times J_h \times K_H}$, $\boldsymbol{\mathcal{Y}}_m \in \mathbb{R}^{I_M \times J_M \times K_m}$ represent the low spatial resolution hyperspectral image (LR-HSI) and MSI, respectively. Let $\boldsymbol{\mathcal{Y}}_s \in \mathbb{R}^{I_M \times J_M \times K_H}$ be the super-resolution

hyperspectral image (SRI) to be estimated. $I_h$, $I_M$ and $J_h$, $J_M$ represent the dimensions of the spatial width and height, respectively, $K_H$, $K_m$ represent the spectral dimension, and $I_h \ll I_M$, $J_h \ll J_M$, $K_m \ll K_H$. It is often assumed that LR-HSI is obtained by spatial degradation of SRI, and MSI is the spectral degradation of SRI, namely,

$$\mathcal{Y}_h = \mathcal{Y}_S \times_1 P_1 \times_2 P_2 \tag{4}$$

$$\mathcal{Y}_m = \mathcal{Y}_S \times_3 P_3 \tag{5}$$

where $P_1 \in \mathbb{R}^{I_h \times I_M}$ and $P_2 \in \mathbb{R}^{J_h \times J_M}$ are the spatial blurring and downsampling matrices, $P_3 \in \mathbb{R}^{K_m \times K_H}$ is a spectral response matrix.

Then, the HSI–MSI fusion can be formulated as

$$\min_{\mathcal{Y}_S} \left|\left| \mathcal{Y}_h - \mathcal{Y}_S \times_1 P_1 \times_2 P_2 \right|\right|_F^2 + \left|\left| \mathcal{Y}_m - \mathcal{Y}_S \times_3 P_3 \right|\right|_F^2 \tag{6}$$

By assuming $\mathcal{Y}_S$ follows the rank-$(L_r, L_r, 1)$ model as shown in (1), the HSI–MSI fusion model (6) can be rewritten as follows:

$$min_{A_r, B_r, C_r} \| \mathcal{Y}_h - \sum_{r=1}^{R} \left( P_1 A_r (P_2 B_r)^T \right) \circ c_r \|_F^2 + \| \mathcal{Y}_m - \sum_{r=1}^{R} \left( A_r B_r^T \right) \circ P_3 c_r \|_F^2 \tag{7}$$

Formula (7) tells us that estimating SRI $\mathcal{Y}_S$ with BTD is equal to estimating the high-resolution abundance map $\left\{ A_r B_r^T \right\}_{r=1}^{R}$ and endmember matrix $C_r$. The theorem in [44,45] gives the recoverability of a hyperspectral super-resolution problem under the BTD of rank-$(L_r, L_r, 1)$.

## 3. Proposed Methods

Generally, the HSI–MSI fusion model (7) is often added with more constraints to achieve a stable and accurate solution, such as total variation, sparsity, low-rank, graph Laplacian, etc. Among them, graph Laplacian can preserve the local manifold structure of high-dimensional image data. Moreover, considering that the superpixel provides local homogeneous regions with geometric structure involved, a superpixel-based graph Laplacian combined with a coupled BTD HSI–MSI fusion model (SGLCBTD) is proposed.

The flowchart is given in Figure 1. As it shows, the core idea of the proposed method is the construction of two graph Laplacian. To achieve this goal, four steps are performed: (1) segmenting the MSI to generate superpixel blocks; (2) extracting the features of the superpixel blocks to form two feature tensors; (3) constructing two graphs based on feature tensors, including compute the weights; and (4) establishing two graph Laplacian terms. Finally, the regularized graph Laplacian terms are introduced to the HSI–MSI fusion framework with the BTD formula.

### 3.1. Superpixel-Based Graph Laplician Construction

3.1.1. Regional Clustering-Based Superpixel Segmentation

Compared with the pixel-wise segmentation method, the superpixel segmentation method shows faster speed and more accurate segmentation results. One of the most popular superpixel segmentation methods is the simple linear iterative clustering (SLIC) algorithm. Inspired by the SLIC method, a regional clustering superpixel segmentation method is designed for HSI unmixing by integrating spatial correlation and spectral similarity at clustering procedure [47]. Specifically, the combination of spectral information divergence (SID) and spectral angler mapper (SAM) is employed as a spectral distance measurement, and Euclidean distance is used as a spatial distance. Taking this merit of the superpixel segmentation method into account, in this paper, the regional clustering superpixel segmentation method is employed to generate superpixel blocks.

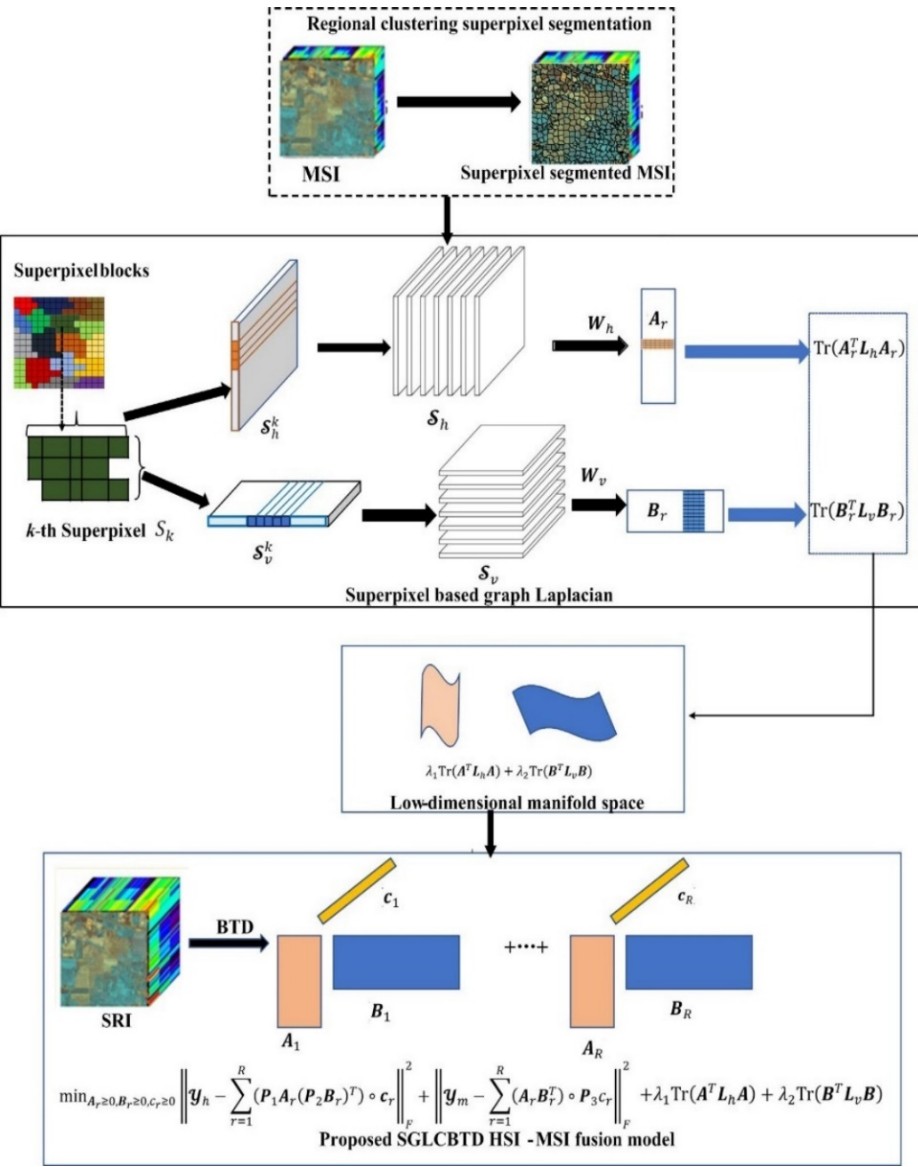

**Figure 1.** Flowchart of the proposed SGLCBTD method.

### 3.1.2. Two-Directional Feature Tensors Extraction

Similarly to the MSI, the resultant superpixel block shows geometric spatial structure as well as spectral information. Along horizontal and vertical spatial directions, the superpixel has different features. Therefore, it is necessary to design two-directional feature tensors to represent the geometric structure.

For each irregular superpixel block $S_k$, along the horizontal direction, feature tensor $\boldsymbol{S}_h^k$ is constructed by its horizontal feature vectors (such as average, maximum, median or difference, etc.). Then, all horizontal feature tensors are arranged according to the second dimension, resulting in a feature tensor $\boldsymbol{S}_h$ with a size of $I_M \times N \times K_m$, where $N$ is the number of the superpixel block. Meanwhile, the vertical feature tensor $\boldsymbol{S}_V \in \mathbb{R}^{I_M \times J_M \times K_m}$ is generated in the same manner.

### 3.1.3. Two Graph Generation

A horizontal graph $G_h = (V_h, E_h)$ with vertex $V_h$ and edge $E_h$ is defined by horizontal feature tensor $\boldsymbol{S}_h$ derived from superpixel blocks. To be specific, taking the one-mode

factorization of tensor $\mathcal{S}_h$ as elements of the vertex $V_h = \left( S_h^{(1)}(1,:), \cdots, S_h^{(1)}(I_M,:) \right)$, the weigh $w_h^{pq}$ of the edge $E_h = \left\{ e_{p,q}^h \right\}$ can be calculated as follows

$$w_h^{pq} = e^{-\frac{\|S_h^{(1)}(p,:) - S_h^{(1)}(q,:)\|^2}{\sigma^2}} \tag{8}$$

where $S_h^{(1)}(p,:)$ and $S_h^{(1)}(q,:)$ represents the $p$-th row and $q$-th row of $S_h^{(1)}$, respectively, and $\sigma$ is the bandwidth of Gaussian kernel.

Meanwhile, a vertical graph = with vertex $V_v$ and edge $E_v$ can be defined according to the vertical feature tensor $\mathcal{S}_v$ in the same way.

### 3.1.4. Two Graph Laplacian Construction

The graph shows the spatial correlation between superpixels, which is the same as that of SRI, for the fused SRI and MSI have the same spatial structure. The manifold structure in MSI is incorporated into SRI as illustrated in Figure 2, where $A_r$ and $B_r$ are the decomposition factor of BTD. With the BTD factorization, the factor matrix $A_r$ represents the spatial information along the horizontal direction (first mode of the tensor), so the row relationship of HR-HSI and MSI is the same as that of $A_r$. Similarly, factor matrix $B_r$ represents the vertical (second mode of tensor) information, so the relationship between columns in HR-HSI and MSI is the same of that of the columns of $B_r$.

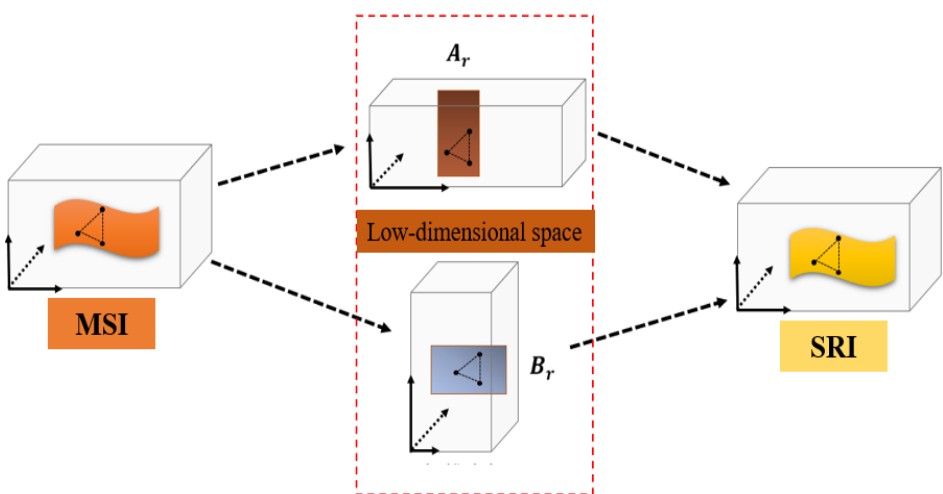

**Figure 2.** The manifold structure preservation between MSI and HR-MSI.

Therefore, the similarity between rows in $A_r$ can be formulated as:

$$\sum_{p=1}^{I_M} \sum_{q=1}^{I_M} \left\| A_r(p,:) - A_r(q,:) \right\|^2 w_h^{pq} \tag{9}$$

where $A_r(p,:)$ and $A_r(q,:)$ represents the $p$-th row and $q$-th row of $A_r$, respectively, $w_h^{pq}$ is the weight of the horizontal graph.

Let $W_h = \left( w_h^{pq} \right) \in \mathbb{R}^{I_M \times I_M}$, which stands for the horizontal weighted adjacency matrix. Then, the horizontal graph Laplacian matrix $L_h$ can be defined as $L_h = D_h - W_h$, where $D_h$ is a diagonal matrix with diagonal elements $d_h^{ii} = \sum_j w_h^{ij}$.

Therefore, Formula (9) can be rewritten as follows:

$$\sum_{p=1}^{I_M} \sum_{q=1}^{I_M} \left\| A_r(p,:) - A_r(q,:) \right\|^2 w_h^{pq} = \text{Tr}\left( A_r^T L_h A_r \right) \tag{10}$$

Extending this constraint to $A = [A_1, \cdots, A_R]$, the horizontal graph Laplacian can be written as

$$\sum_{r=1}^{R} \text{Tr}\left(A_r^T L_h A_r\right) = \text{Tr}\left(A^T L_h A\right) \tag{11}$$

In the same way, the vertical graph Laplacian related to $B_r$ is defined as

$$\sum_{r=1}^{R} \text{Tr}\left(B_r^T L_v B_r\right) = \text{Tr}\left(B^T L_v B\right) \tag{12}$$

where $L_v$ is the vertical graph Laplacian matrix, $B = [B_1, \cdots, B_R]$.

### 3.2. Proposed SGLCBTD Model and Algorithm

Incorporating the two mentioned graph Laplacian terms (11–12) with the coupled BTD super-resolution model (7), a superpixel graph Laplacian regularization with coupled BTD fusion model is proposed (SGLCBTD):

$$\begin{aligned} \min_{A_r \geq 0, B_r \geq 0, c_r \geq 0} \|\mathcal{Y}_h - \sum_{r=1}^{R} \left(P_1 A_r (P_2 B_r)^T\right) \circ c_r\|_F^2 + \|\mathcal{Y}_m - \sum_{r=1}^{R} (A_r B_r^T) \circ \\ P_3 c_r\|_F^2 + \lambda_1 \text{Tr}\left(A^T L_h A\right) + \lambda_2 \text{Tr}\left(B^T L_v B\right) \end{aligned} \tag{13}$$

where $\lambda_1$ and $\lambda_2$ are regularization parameters.

Let us define the objective function in Equation (13) as $J(A, B, C)$, the above super-resolution model is solved alternately by block coordinate descent (BCD) algorithm [48], i.e., the matrices $A, B, C$ are iteratively updated via solving subproblems w.r.t. while fixing other variables as follows

$$A_{t+1} \leftarrow \text{argmin}_{A \geq 0} J\left(A, B^t, C^t\right) \tag{14a}$$

$$B_{t+1} \leftarrow \text{argmin}_{B \geq 0} J\left(A^{t+1}, B, C^t\right) \tag{14b}$$

$$C_{t+1} \leftarrow \text{argmin}_{C \geq 0} J\left(A^{t+1}, B^{t+1}, C\right) \tag{14c}$$

where $t$ is the number of current iteration steps.

Each subproblem is a quadratic optimization problem, which leads to the generalized Sylvester equation and can be transformed into a large-scale sparse linear system of equations by Kronecker product.

Consider the subproblem (14a), fix $B$ and $C$, the subproblem of $A$ can be rewritten as

$$\min_{A \geq 0} \left\| Y_h^{(1)} - (C \odot_p P_2 B)(P_1 A)^T \right\|_F^2 + \left\| Y_m^{(1)} - (P_3 C \odot_p B) A^T \right\|_F^2 + \lambda_1 \text{Tr}\left(A^T L_A A\right) \tag{15}$$

The optimization problem is quadratic, and its solution is equivalent to compute the following general Sylvester equation [49]:

$$P_1^T P_1 A Q^T Q + A K^T K + 2\lambda_1 L_A A = P_1^T Y_h^{(1)T} Q + Y_m^{(1)T} K \tag{16}$$

where $Q = C \odot_p P_2 B, K = P_3 C \odot_p B$.

For the solution of subproblem of matrices $B$ and $C$, the algorithms are the same as that of matrix $A$.

## 4. Experimental Results

### 4.1. Experiment Setup

In this section, to demonstrate the effectiveness of the proposed HSI super-resolution method, numerical experiments are carried out on two popular datasets: Indian Pines and Pavia University datasets (https://www.ehu.eus/ccwintco/index.php/Hyperspectral_Remote_Sensing_Scenes, accessed on 20 March 2022). Both qualitative and quantitative analysis are used in our experiments.

4.1.1. Quality Assessment Indices

For quantitative performance, seven indices are employed to evaluate the performance [7].

(1)　Normalized mean square error *(NMSE)* is defined as

$$NMSE = \frac{\left|\left|\boldsymbol{\mathcal{Y}} - \hat{\boldsymbol{\mathcal{Y}}}\right|\right|_F^2}{\left|\left|\boldsymbol{\mathcal{Y}}\right|\right|_F^2} \tag{17}$$

where $\boldsymbol{\mathcal{Y}} \in \mathbb{R}^{I \times J \times K}$ is the ideal HSI and $\hat{\boldsymbol{\mathcal{Y}}} \in \mathbb{R}^{I \times J \times K}$ is the resulted SRI.

(2)　Reconstruction signal-to-noise ratio (*R-SNR*) is inversely proportional to NMSE with the formulation as follows:

$$R\text{-}SNR = 10\lg\left(\frac{\left|\left|\boldsymbol{\mathcal{Y}}\right|\right|_F^2}{\left|\left|\boldsymbol{\mathcal{Y}} - \boldsymbol{\mathcal{Y}}\right|\right|_F^2}\right) \tag{18}$$

(3)　Spectral angle mapper (*SAM*) evaluates the spectral distortion and is defined as

$$SAM = \frac{1}{IJ} \sum_{n=1}^{IJ} \arccos\left(\frac{\boldsymbol{Y}^{(3)}(n,:), \hat{\boldsymbol{Y}^{(3)}}(n,:)}{\left|\left|\boldsymbol{Y}^{(3)}(n,:)\right|\right|_2 \cdot \left|\left|\hat{\boldsymbol{Y}^{(3)}}(n,:)^T\right|\right|_2}\right) \tag{19}$$

where $\boldsymbol{Y}^{(3)}(n,:)$ is the spectral vector, $\langle . \rangle$ is inner product of two vectors.

(4)　Relative global dimensional synthesis error (*ERGAS*) reflects the global quality of the fused results and is defined as

$$ERGAS = 100d\sqrt{\frac{1}{K} \sum_{k=1}^{K} \frac{\left|\boldsymbol{\mathcal{Y}}(:,:,k) - \hat{\boldsymbol{\mathcal{Y}}}(:,:,k)\right|_F^2}{\mu_k^2}} \tag{20}$$

where $d$ is spatial upsampling factor, $\mu_k$ is the mean of $\hat{\boldsymbol{\mathcal{Y}}}$.

(5)　Correlation coefficient (*CC*) is computed as follows

$$CC = \frac{1}{K} \sum_{k=1}^{K} \rho\left(\boldsymbol{\mathcal{Y}}(:,:,k), \hat{\boldsymbol{\mathcal{Y}}}(:,:,k)\right) \tag{21}$$

where $\rho$ is the Pearson correlation coefficient.

(6)　Peak signal-to-noise rate (*PSNR*) for each band of HSI is defined as

$$PSNR = 10\lg\left(\frac{\max(\left|\left|\boldsymbol{\mathcal{Y}}(:,:,k)\right|\right|_F^2)}{\left|\left|\boldsymbol{\mathcal{Y}}(:,:,k) - \hat{\boldsymbol{\mathcal{Y}}}(:,:,k)\right|\right|_F^2}\right) \tag{22}$$

(7)　Structural similarity index measurement (*SSIM*) for each band of HSI is defined as follows:

$$SSIM = \frac{\left(2\mu_{\boldsymbol{\mathcal{Y}}}\mu_{\hat{\boldsymbol{\mathcal{Y}}}} + c_1\right)(2\sigma + c_2)}{\left(\mu_{\boldsymbol{\mathcal{Y}}}^2 + \mu_{\hat{\boldsymbol{\mathcal{Y}}}}^2 + c_1\right)\left(\sigma_{\boldsymbol{\mathcal{Y}}}^2 + \sigma_{\hat{\boldsymbol{\mathcal{Y}}}}^2 + c_2\right)} \tag{23}$$

where $\mu_1, \mu_2$ and $\sigma_1^2, \sigma_2^2$ are the mean and variance of the k-th band image $\boldsymbol{\mathcal{Y}}(:,:,k)$ and $\hat{\boldsymbol{\mathcal{Y}}}(:,:,k)$, respectively, $\sigma$ is the covariance between $\boldsymbol{\mathcal{Y}}(:,:,k)$ and $\hat{\boldsymbol{\mathcal{Y}}}(:,:,k)$, and $c_1, c_2$ are constant.

4.1.2. Methods for Comparison

The proposed SGLCBTD method is compared with the state-of-the-art HSI–MSI fusion methods, including: coupled nonnegative matrix factorization (CNMF) [3], super-resolution tensor-reconstruction (STEREO) [37], STEREO with non-negative CP decomposition (CNN-CPD) [37], coupled non-negative tensor block term decomposition (CNN-BTD) [43], and graph Laplacian regularization with coupled block term decomposition (GLCBTD) [46].

In the experiment, the optimal parameters involved in different methods are set according to the author's suggestion. Specially, in tensor-based method, considering that the Pavia University dataset has rich spatial structures, the rank of tensor $R$ is set to 15, but for the Indian Pines dataset, the rank is 10. Meanwhile, in the related BTD rank-$(L_r, L_r, 1)$ methods, including CNN-BTD, GLCBTD and SGLCBTD, the rank of the factor matrices is $L_r = 10$. In addition, regularization parameters $\lambda_1$ and $\lambda_2$ in SGLCBTD are set to $10^{-5}$. More details about the parameters are shown in Section 4.3.

*4.2. Performance Comparison of Different Methods*
4.2.1. Indian Pines Dataset

The Indian Pines dataset was captured by the NASA AVIRIS instrument. The size of the underlying SRI $\mathcal{Y}_S$ is $145 \times 145 \times 220$ and the wavelength covers 400–2500 nm. The HSI and MSI are obtained according to the Wald protocol [50]. The degradation from SRI to HSI is that blurring with $9 \times 9$ Gaussian and downsampling with factor 5, resulting in $\mathcal{Y}_h$ with size of $29 \times 29 \times 220$. The size of the MSI is $145 \times 145 \times 4$. Finally, zero-mean iid. Gaussian noise is added to HSI and MSI, and the signal-to-noise ratio (SNR) is set to 30 dB.

Table 1 lists the values of R-SNR, NMSE, SAM, ERGAS and CC of the compared methods on the Indian Pines dataset, and the best values are marked in bold. It can be shown that the tensor-based methods outplay matrix-based methods. The indexes of BTD fusion methods are much better than that of CPD. Moreover, the regularized models, CNN-CPD, GLCBTD and SGLCBTD are more advantageous than STEREO and CNN-BTD methods, for the latter two have rather low indices. Most of indices of the proposed SGLCBTD achieve first place, which shows that the method performs well in spatial detail and spectral preservation. In addition, compared with the GLCBTD, the R-SNR is significantly improved by about 1.76 dB.

**Table 1.** Quantitative indices comparison on the Indian Pines dataset.

| Algorithm | R-SNR | NMSE | SAM | ERGAS | CC |
|:---:|:---:|:---:|:---:|:---:|:---:|
| CNMF | 25.24 | 0.0546 | 0.0398 | 1.3320 | 0.7589 |
| STEREO | 26.25 | 0.0487 | 0.0413 | 1.1254 | 0.7970 |
| CNN-CPD | 26.41 | 0.0478 | 0.0365 | 1.0891 | 0.7980 |
| CNN-BTD | 27.32 | 0.0430 | 0.0346 | 1.0394 | 0.8070 |
| GLCBTD | 28.02 | 0.0397 | 0.0319 | **0.9507** | 0.8193 |
| SGLCBTD | **29.78** | **0.0324** | **0.0277** | 0.9610 | **0.8128** |

The PSNR and SSIM values for each spectral band of various fusion methods are compared in Figure 3. As it can be seen, most of CNMF's curves achieve the lowest values in each band, which means the performance of the matrix-based method is significantly lower than that of tensor-based methods. The curves of SREREO and CNN-CPD are very similar except for a higher SSIM index of CNN-CPD. Three BTD-based methods, CNN-BTD, GLCBTD and SGLCBTD perform better than other methods. The overall performances of GLCBTD and SGLCBTD are excellent, especially in SSIM curves, which shows that the graph-based method can preserve the geometric structure well. It should be mentioned, in the PSNR curve, from the 100-th band, the curve of SGLCBTD has unsatisfactory results, which is likely due to the water absorption, noise and parameters' setting in our view. Even so, the curve is the most stable in each band of all compared methods. In general, the

PSNR and SSIM curves show that the proposed SGLCBTD method has the superiority of preserving geometric structures.

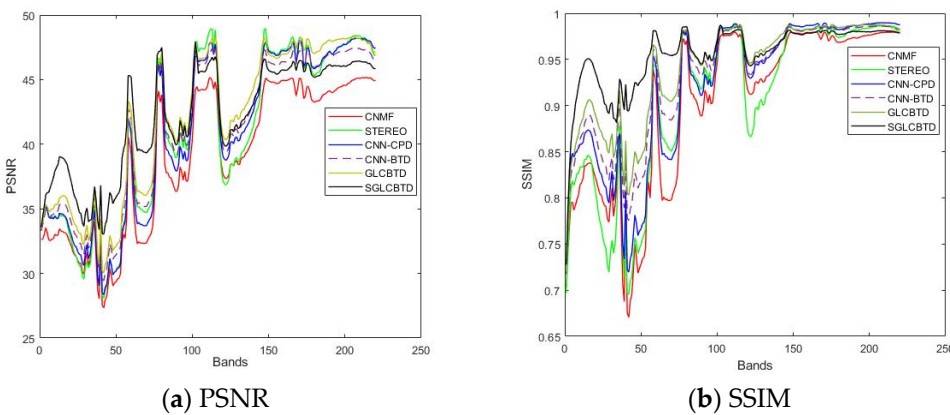

|  |  |
| :---: | :---: |
| (**a**) PSNR | (**b**) SSIM |

**Figure 3.** PSNR and SSIM in function of each band for different methods on the Indian Pines dataset.

Figures 4 and 5 show the fusion results on the Indian Pines dataset in 10-th and 100-th bands, respectively. From the visual effect, it can be seen that SGLCBTD is closer to the original image with reduced red areas and increased blue areas, moreover, the edge details are clearer than those of other methods.

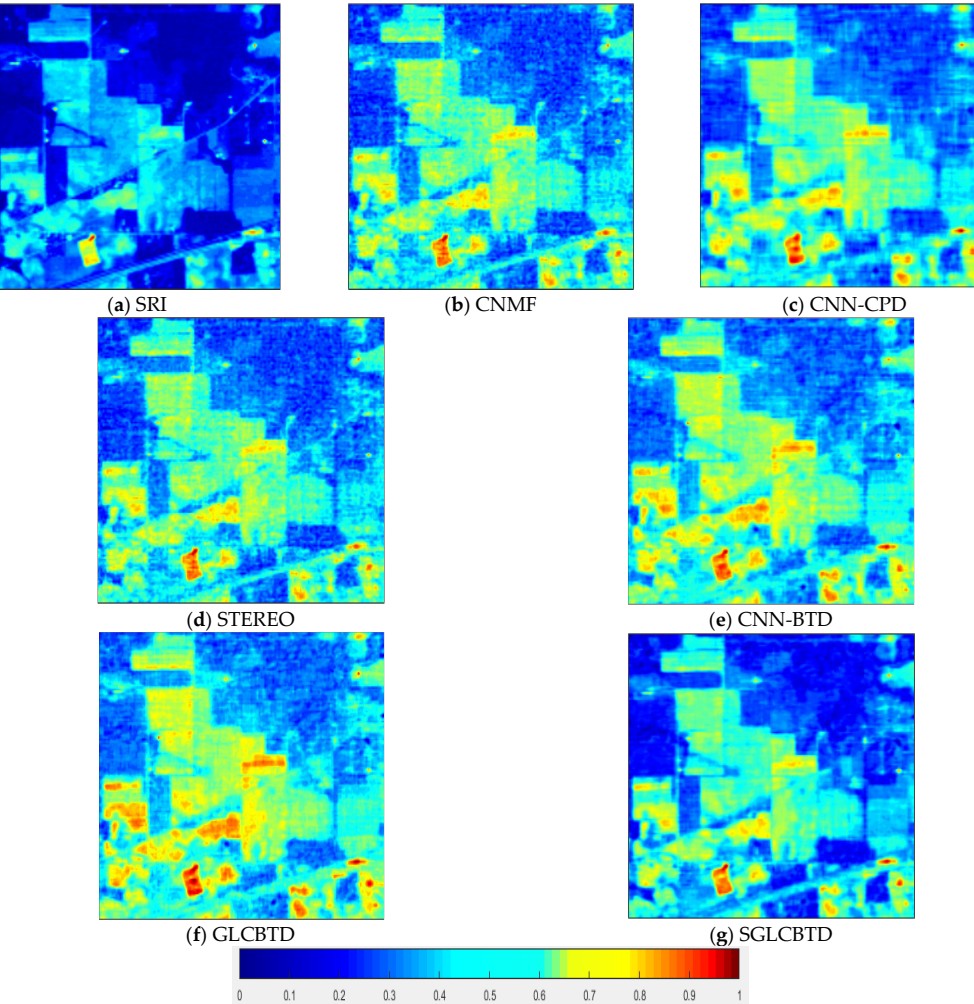

**Figure 4.** Comparisons of fusion results of the 10-th band on the Indian Pines dataset.

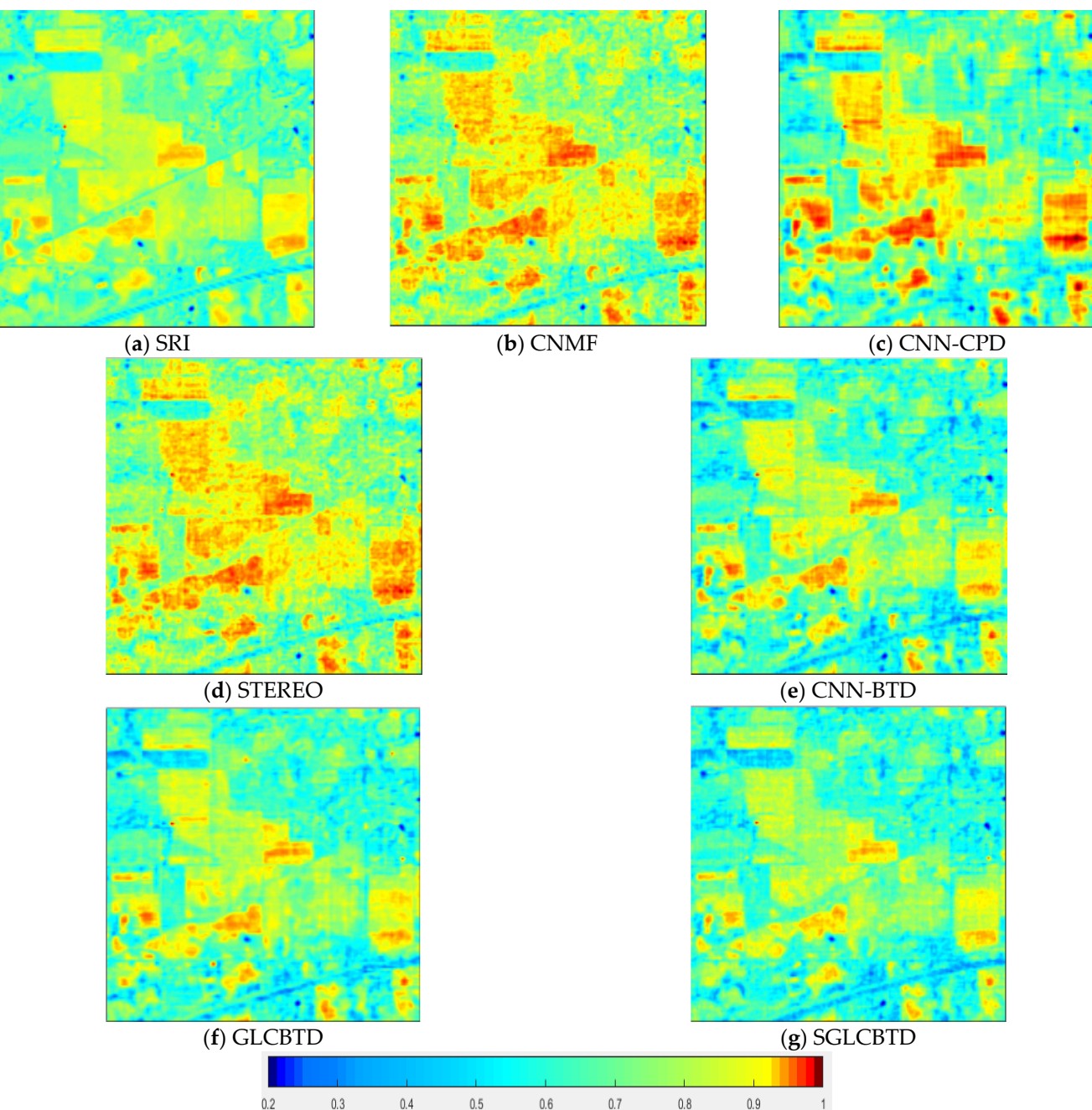

**Figure 5.** Comparisons of fusion results of the 100-th band on the Indian Pines dataset.

### 4.2.2. Pavia University Dataset

The Pavia University dataset was captured by the ROSIS instrument. The size of the original HSI is $610 \times 340 \times 115$, where the spectral band is from 430 nm to 860 nm. The spatial size of $200 \times 200$ is cropped due to hardware limitations, and the spectral band is 103 after removing the vapor absorption bands. Therefore, the underlying SRI $\mathcal{Y}_s$ is $200 \times 200 \times 103$. A $9 \times 9$ Gaussian blurring and downsampling factor 4 are performed as degradation to obtain a HSI $\mathcal{Y}_h$ with size of $50 \times 50 \times 103$. The size of MSI $\mathcal{Y}_m$ is $200 \times 200 \times 4$. Finally, zero mean iid. Gaussian noise is added to HSI and MSI, the signal-to-noise ratio (SNR) is set to 25 dB.

Table 2 shows the numerical results of several comparison methods on the Pavia University dataset. It can be seen that for the Pavia University dataset with tensor rank-10, the fusion performance fusion of CNN-BTD and GLCBTD have been improved. GLCBTD

is still better than CNN-BTD due to the regularization term, while SGLCBTD improves the R-SNR value of GLCBTD by 0.3 dB, and other indices have also been improved, which shows that the proposed SGLCBTD method can improve the spatial–spectral assessment values effectively.

**Table 2.** Quantitative indices comparison on the Pavia University dataset.

| Algorithm | R-SNR | NMSE | SAM | ERGAS | CC |
|:---------:|:-----:|:----:|:---:|:-----:|:--:|
| CNMF | 17.36 | 0.1355 | 0.1105 | 1.0453 | 0.9526 |
| STEREO | 18.21 | 0.1228 | 0.1465 | 1.0154 | 0.9624 |
| CNN-CPD | 18.75 | 0.1154 | 0.1052 | 0.8867 | 0.9652 |
| CNN-BTD | 19.41 | 0.1070 | 0.1042 | 0.8447 | 0.9702 |
| GLCBTD | 21.08 | 0.0883 | 0.0873 | 0.7100 | 0.9792 |
| SGLCBTD | **21.38** | **0.0853** | **0.0855** | **0.6919** | **0.9807** |

Figure 6 shows the comparison of PSNR and SSIM curves on the Pavia University dataset of different fusion methods, in which the red solid line represents the proposed SGLCBTD method. It can be seen that the PSNR and SSIM curves of GLCBTD and SGLCBTD are roughly similar, and both of them have higher values than that of other methods. Furthermore, it should be noticed that the overall curves of SGLCBTD are the highest and most stable, which means the proposed SGLCBTD performs best among the compared methods.

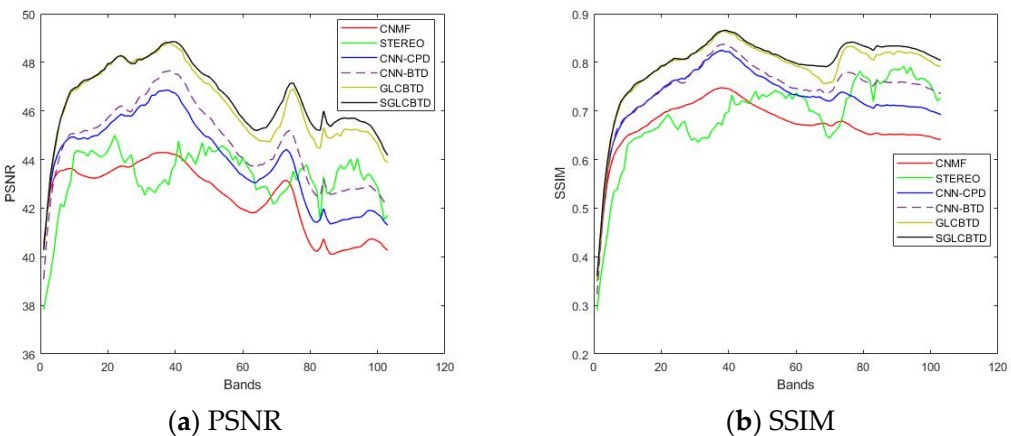

(**a**) PSNR      (**b**) SSIM

**Figure 6.** PSNR and SSIM in function of each band for different methods on the Indian Pines dataset.

As shown in Figures 7 and 8, the 10-th band and the 80-th band of the fusion results on the Pavia University dataset image are displayed, respectively. CNMF, CNN-CPD and STEREO have more noise unremoved as well as more blur. Relatively, BTD-based methods achieve better fusion results, among them, regularized GLCBTD and SGLCBTD methods achieve better visual effects than that of CNN-BTD. In addition, the spatial edge and texture structures of the fused image of SGLCBTD is clearer with less noise, which shows the effectiveness of SGLCBTD fusion method.

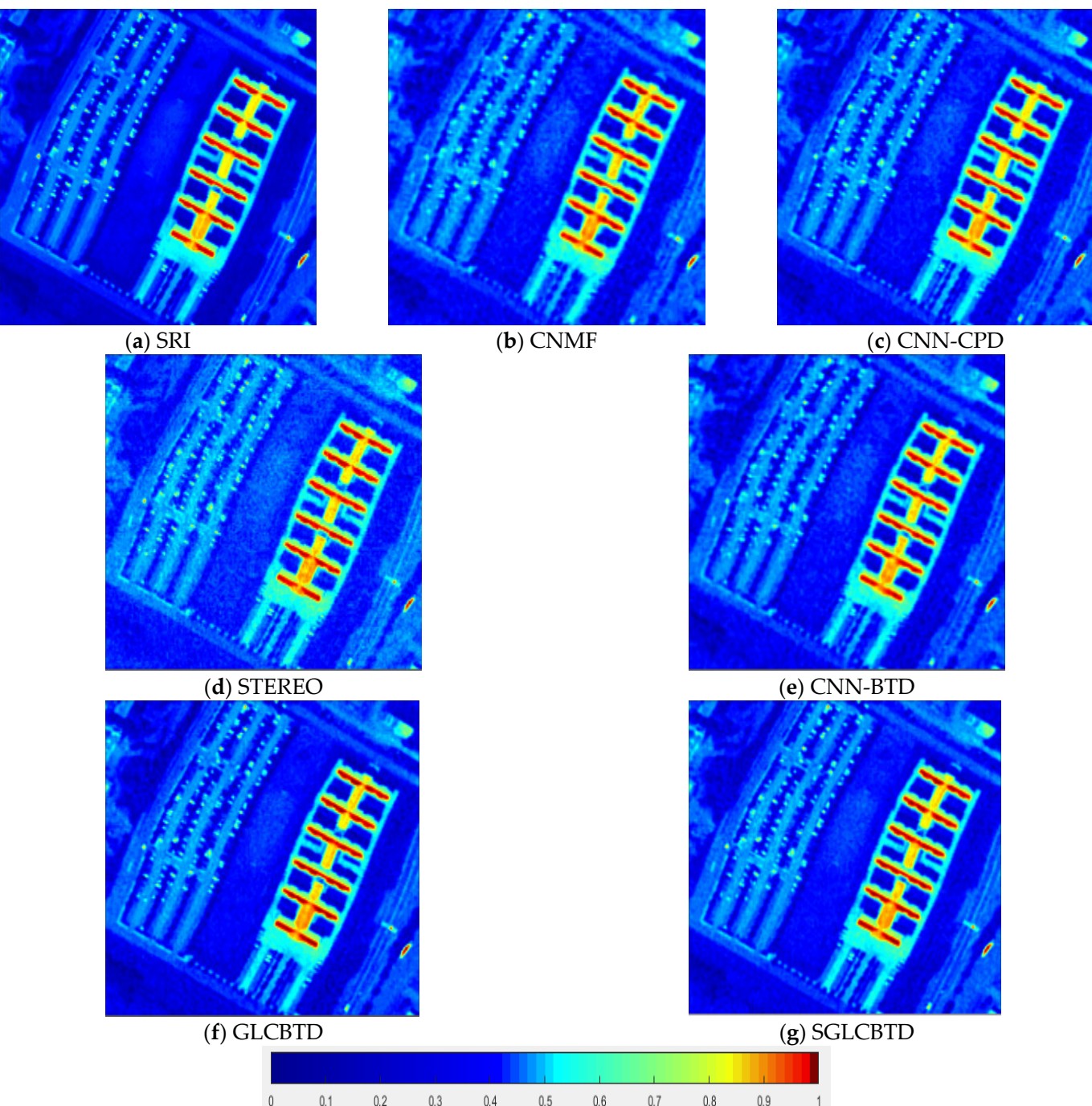

**Figure 7.** Comparisons of fusion results of the 10-th band on the Pavia University dataset.

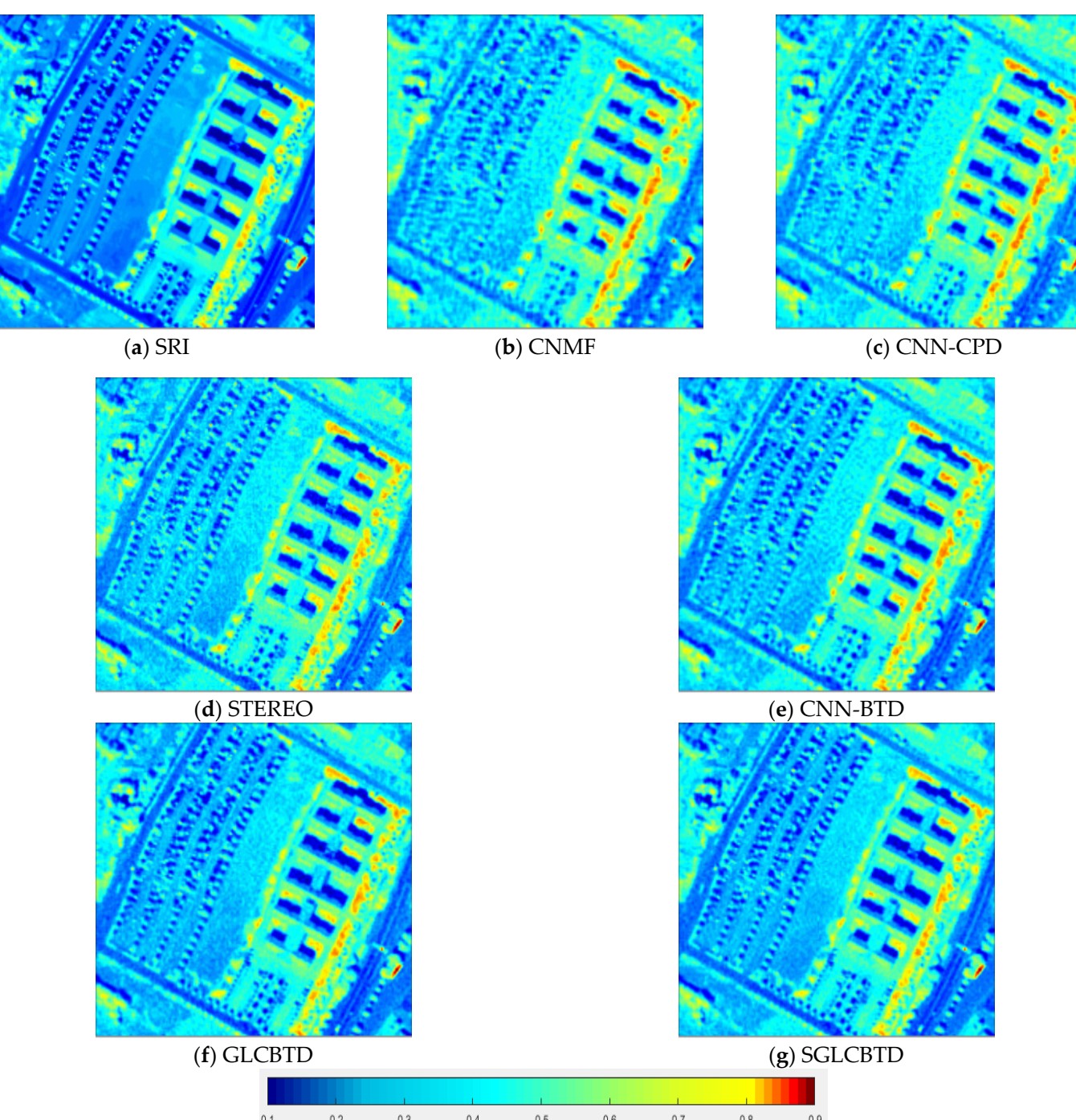

**Figure 8.** Comparisons of the fusion results of the 80-th band on the Pavia University dataset.

*4.3. Discussions*

4.3.1. Parameter Analysis

The parameters involved in the SGLCBTD including the number of superpixel $N$, the tensor rank $R$, the rank of factor matrices $L$, and $\lambda_1, \lambda_2$. The main contribution of this paper is the tensor BTD with a superpixel-based graph. Therefore, the parameter analysis is performed on $R$, $L$ and $N$.

(1)　Analysis of $R$ and $L$

Taking the Indian Pines dataset as an example, R-SNR is used to evaluate the performance of parameters $R$ and $L$ as shown in Figure 9. Given the range of $R$ and $L$ form

7–13 with increment 1. It can be seen that both of two curves reach the highest peak values around 10. In addition, with $R$ or $L$ decrease, the performance becomes worse. Considering the high computational complexity with high rank increase, for the Indian Pines, $R = L = 10$, while for the Pavia University dataset, $R = 15$, $L = 10$ on account of the complex geometric structures in the image.

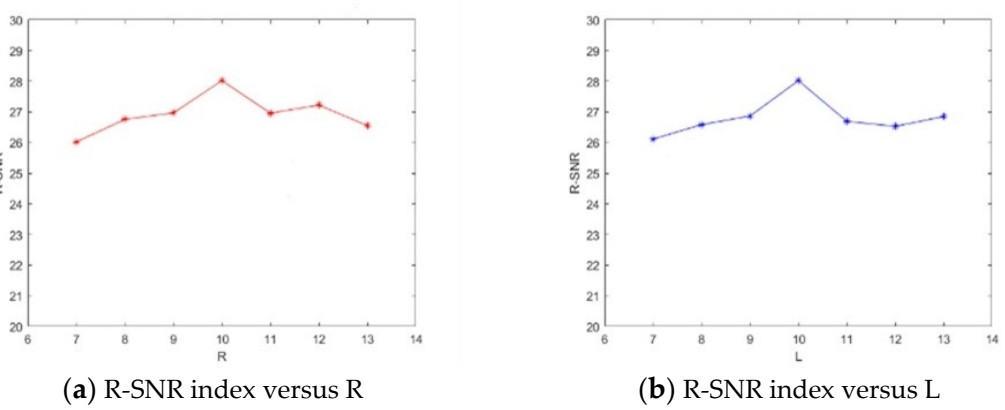

(**a**) R-SNR index versus R          (**b**) R-SNR index versus L

**Figure 9.** R-SNR varies with different R and L on the Pavia University dataset.

(2)    Analysis of *N*

The proposed SGLCBTD method is a superpixel-based method, the number of superpixel $N$ is crucial to the efficient as well as effectiveness. For two datasets, the parameter N is selected from the set {225, 324, 400, 625, 841, 900} and {225, 324, 400, 484, 625,784}, respectively, by experience. Figure 10 shows R-SNR curves as parameter N changes. From the Figure, for the Indian Pines dataset, R-SNR reaches the highest value when $N = 400$. When $N > 400$, the curve drops down then rises at the point $N = 625$. For the Pavia University dataset, R-SNR is relatively stable in the data range. In addition, when $N \in [400, 700]$, the highest value is reached at $N = 625$. Compared to the Indian Pines dataset, the Pavia University dataset has a richer geometry, and the segmentation results should be finer, resulting in a larger number of superpixel blocks. It is also noted that large superpixel numbers increase the time computation and do not lead to better performance. Therefore, 400 and 625 are set as superpixel numbers for the Indian Pines and Pavia University datasets, respectively. More adaptive and accurate estimation of the parameter is still an open issue to be researched further.

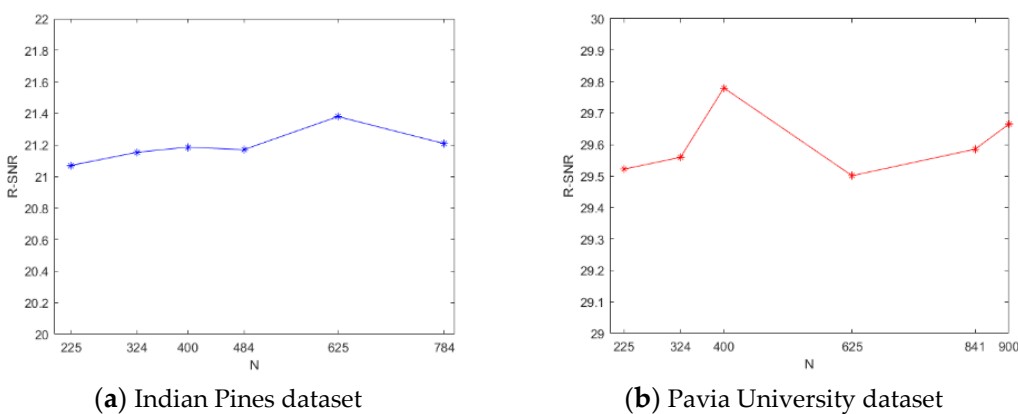

(**a**) Indian Pines dataset          (**b**) Pavia University dataset

**Figure 10.** R-SNR varies with different *N* on two datasets.

4.3.2. Time Complexity Analysis

The computational time of the compared methods on the two datasets is listed in Table 3. The running time results are recorded in MATLAB R2018b, using a GPU server with

NVIDIA RTX 2080Ti/11GB. Owing to the advantages of the block coordinate descent (BCD) algorithm or the alternating direction method of multipliers (ADMM), all the compared methods converge quickly after 10 to 25 iterations. In addition, for each dataset, the time given in Table 3 is obtained by averaging the five times. As can be seen, the STEREO is the fastest method. GLCBTD and the proposed SGLCBTD methods cost more time than those of other methods due to the massive calculation of superpixel segmentation and adjacency matrix of the graph, which is a progressive work to be considered for us in the future.

**Table 3.** Computational time of the compared methods (seconds).

| Method | Indian Pines Dataset | Pavia University Dataset |
|---|---|---|
| CNMF | 9.02 | 11.34 |
| STEREO | **2.97** | **4.76** |
| CNN-CPD | 5.05 | 5.60 |
| CNN-BTD | 65 | 89 |
| GLCBTD | 534 | 771 |
| SGLCBTD | 1023 | 1245 |

## 5. Conclusions

In this paper, a HSI super-resolution method is proposed based on tenor block term decomposition, known as SGLCBTD. To preserve the spatial manifold structure of the fused HSI, two-directional spectral–spatial graphs are constructed according to feature tensors induced by the MSI segmented superpixel. Then, the manifold graph Laplacian is utilized to regularize the super-resolution HSI, resulting in the proposed HSI–MSI fusion method. In addition, the model is solved alternately by block coordinate descent algorithm. Stimulation experiments are conducted on different datasets. Compared with the state-of-the-art methods, the proposed SGLCBTD obtains better fusion performance with more spatial details retained.

For future work, two aspects should be taken into consideration. On one hand, the feature of superpixel blocks is limited, more powerful features can be exploited to further improve the preservation of the manifold structures of the HSI. On the other hand, as mentioned in Section 4.3.2, more efficient optimization algorithms need to be developed to reduce time complexity.

**Author Contributions:** Methodology, writing-review and editing, H.L.; Conceptualization, writing—original draft preparation, software, W.J.; Validation, data curation, Y.Z.; Supervision, Z.W. All authors have read and agreed to the published version of the manuscript.

**Funding:** This research was funded in part by the National Natural Science Foundation of China under Grant 61971223 and Grant 61871226, in part by the Fundamental Research Funds for the Central Universities under Grant 30917015104, and in part by the Fundamental Research Funds for the Central Universities under Grant JSGP202204.

**Data Availability Statement:** The dataset use in this paper can be found form the link: https://www.ehu.eus/ccwintco/index.php/Hyperspectral_Remote_Sensing_Scenes accessed on 20 March 2022).

**Conflicts of Interest:** The authors declare no conflict of interest.

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
