# Peer review of "Coupled Tensor Block Term Decomposition with Superpixel-Based Graph Laplacian Regularization for Hyperspectral Super-Resolution"

_remotesensing, doi:10.3390/rs14184520_

Round 1

Reviewer 1 Report

This paper proposes a superpixel based graph Laplacian regularized tensor coupled block-term decomposition (BTD) super-resolution model (SGLCBTD) for the hyperspectral image (HSI) super-resolution. Based on the coupled block-term decomposition model, the superpixel graph is constructed by two superpixel feature tensors related to two factor matrices. Furthermore, the spatial manifold structure of the multispectral image (MSI) is transferred to HR-HSI by the graph Laplacian regularization, leading to an improvement of the spatial preservation in the fused image. Experimental results on two datasets demonstrate that the proposed SGLCBTD method achieves better fused results compared with other advanced super-resolution methods, especially on the improvement of the spatial structure.

The main contributions of the paper are as follows:

(1) It proposes two directional tensor graphs via the features of the segmented MSI superpixel blocks, whose local geometric structure is consistent with HSI;

(2) The spatial similarity weights of the superpixel blocks are calculated and graph Laplacian matrices are constructed, which are used to convey the spatial manifold structure from MSI to the factor matrices of HSI;

(3) The proposed superpixel graph Laplacian BTD model is solved by the block coordinate descent algorithm, which achieves better fused results compared with other advanced super-resolution methods.

The proposed method in this paper is effective and it is a topic of interest in the related areas. I have some minor comments as follows:

(1) There are some expressions in the paper that do not use a uniform format, such as ‘super resolution’ and ‘super-resolution’, ‘data set’ and ‘dataset’, ‘with size of’ and ‘with size’;

(2) There are also some tense errors. For example, in the last line of the ‘conclusions’ part, ‘obtains’ should be used instead of ‘obtain’;

(3) The colors of the curves in figure 3 and figure 6 are slightly different from each other and easy to be confused, which need to be adjusted;

(4) In the figure 3(a), when the horizontal ordinate is greater than 100, the position of the SGLCBTD curve is very bad. I hope the reason for this result can be analyzed.

Author Response

The authors would like to convey our sincere thanks to the reviewer for your valuable time and comments to improve the quality of this manuscript. Your suggestions helped us in improving the content of the paper significantly. A point-wise response to the comments is attached. The response to the comments is highlighted with yellow color in the revised version.

Reviewer 2 Report

My comments:

1. The topic of this paper is interesting and it will contribute in related research field.

2. A section of “Related Works” or “Literature Review” is necessary for this paper.

3. A separate section of Discussion” before “Conclusions” is necessary for this paper.

4. The section “Conclusion” must be reinforced more. For example, the more contributions to academic research as well as theoretical implications, research limitations, and suggestions for further research.

Author Response

(The authors gave the same response as above.)

Reviewer 3 Report

This paper describes a new method to compute high-resolution hyperspectral images using low-resolution hyperspectral images and high-resolution multispectral images. The strong point of the manuscript is its solid and detailed mathematical foundations, which are explained extensively in the paper. However, the experimentation section is very poor. For example, the paper does not indicate how many images were used for testing the proposed method, and the study of the computational efficiency is completely obviated. Although the manuscript is well-organized, the writing is deficient, so the text should be completely revised to correct writing errors and weird expressions.

These are my specific comments:

1. The abstract is very difficult to understand. Only after reading the paper, the reader can understand the abstract. However, the purpose of the abstract should be to have an idea of the paper before reading it. I know that the proposed method is very complex. But you should do an effort to try to explain it more easily. For example, you indicate that the method is "a superpixel based graph Laplacian regularized tensor coupled block term decomposition (BTD) super-resolution model". Instead of that long sentence, you can describe the method step-by-step: first, superpixels are computed for the MSI image; then a graph is created using distance between features of the superpixels; etc. Also, you can add some numerical results of the proposed method in the abstract.

2. It is a pity that you have not numbered the lines of the paper, since they would be very useful to refer to specific lines of the document. You should number them in the revised version.

3. Before "(" and "[", you must always write a blank apace. This applies to the citations, definition of acronym, and all the uses of these symbols. For example, instead of "decomposition(CPD)[30]", it should be "decomposition (CPD) [30]". Please, revise all the paper.

4. As indicated in point 1, the mathematical basis to solve this problem is very complex, and can be difficult to understand for an average reader. This also applies to the introduction, which presents many complex concepts with no explanation. You should do an effort to make the introduction easier to read and understand, briefly explaining the main mathematical terms that are introduced.

5. In the introduction, you say "Tensor is a multi-dimension array". But, in your case, you always use up to 3D arrays, is it correct?

6. In section 2, in the definition of the Khatri-Rao product, what does K and p mean? What is the definition of this type of product?

7. Figure 1 should be placed after it is mentioned in the text, not before. The same with Figure 7.

8. In section 3.1, what superpixel algorithm was used to create the superpixel segmentation? It is not mentioned in the paper.

9. The caption of Figure 2 should be improved. You must define all the elements that are used in this figure.

10. In equation (8), is it sigma, or should it be sigma^2?

11. In section 4, you should provide a reference for the two datasets used in the experiments. What is the size of these datasets (number of images)? How many images were used for training and for testing?

12. "Pavia University" is a university, not a dataset of images. You should say "Pavia University dataset", not just "Pavia University". Or, in short, you can say "PU dataset", and "IP dataset". Please, revise all the mentions to these datasets in the paper.

13. The evaluation indices used in the experiments should be defined. For example, I recommend you add a new table including the definition of these indices.

14. In the list of methods used for comparison (in page 8), you should use the format of the journal for numbered lists. This list is not in the correct format. Also, where did you obtain the implementation of these methods? Are they your own implementations or are they publicly available? This should be explained in the manuscript.

15. In section 4.1, the sentence "which shows that the strong ability of the spatial detail preservation well as well as the robustness" has a very weird expression. It should be rewritten with a clearer expression.

16. The captions of Figures 3, 4 ... 8 should be improved. They must be more informative about the elements contained in the graphs. For example, the caption of Figure 3 can be: "Results of the evaluation criteria for each spectral band, for the proposed method (SGLCBTD) and the methods used for comparison. (a) Peak signal to noise rate (PSNR). (b) Structural similarity index (SSIM)."

17. In Figures 4, 5, 7 and 8, you should include a graphical scale indicating the meaning of the color values.

18. What spectral wavelengths correspond to the 10-th, 80-th and 100-th bands shown in the images?

19. In page 9, you say "Figure 4 and Figure 5 are fusion images in the form of two-dimensional contour map...". A "contour map" is an image which represents the contours of another image. However, the images in Figures 4 and 5 are not contour maps. They are just 2D images.

20. In section 4.2, you say "A subimage of Pavia University dataset with size 200×200×103 is used as SRI Ys". Does it mean that you only used one image in the experiments? This seems to be a writing error. Because, if you have only used one image for the experiments, that is clearly insufficient, and you should repeat the experiment with many more test images.

21. Section 4 ends very abruptly. You present the results of the experiments, and then we have the conclusions. There is no discussion of the obtained results. You don't analyze in detail the obtained results. The discussion of the advantages and disadvantages of the proposed method is missing. What are the weak points of your method?

22. Also, you don't say anything regarding the computational efficiency of your method and the other methods used for comparison. You should report the time required by all the methods during the training stage, and during the test stage. You should indicate the computer and execution environment, and then you should discuss the results with respect to the computational efficiency.

23. The conclusions are very poor and should be improved. Your conclusions only contain a brief description of the method. You should present the most interesting aspects of your research, the possible applications of your methods, the weak points that are pending to be improved, and the future lines of research.

24. The writing of the manuscript should be completely revised, since there are many spelling, syntax and grammatical errors, and weird expressions. For example, here is a list of some errors (but this list is not complete):

S1. super-solution -> super-resolution

S1. depended on -> depend on

S1. Remove the blank line in the second page

S1. represent forms -> representation forms

S1. is well-known -> is a well-known

S1. HR-HSI and MSI were shared the same -> HR-HSI and MSI share the same

S1. These literatures -> These works/papers

S2. resulting in the following -> results in the following

S3. (SGLCBTD)) -> (SGLCBTD)

Figure 1. Superpixe -> Superpixel

S3.1. HR_HSI -> HR-HSI

S3.2. is the regularization parameter -> are the regularization parameters.

S3.2. Let the objective -> Let define/denote the objective

S3.2. The subproblem of A -> the subproblem of A

S3.2. is quadratics -> is quadratic

S4. indexes -> indices

S4. Pavia university -> Pavia University

S4.1. outplay -> outperform

S4.1. are show more -> are more

S4.1. STREEO -> STEREO

S4.1. details are clearer than that of -> details are clearer than those of

S4.2. values than that of -> values than those of

S5. obtain -> obtains

Author Response

(The authors gave the same response as above.)

Round 2

Reviewer 3 Report

The authors have correctly addressed all the questions and comments suggested in my review, doing the necessary changes. Consequently, the manuscript has significantly improved. However, there are still two points that should be corrected:

1. The color scale that you have used in figures 4, 5, 7 and 8 is clearly incorrect. For example, many pixels of the images have red color, but the scale does not contain red color. Moreover, it can be seen in the images that the scale is: blue - cyan - green - yellow - red. However, your color scale contains: blue - cyan - green - orange - yellow; and the blue-cyan colors go from 0 to 0.5, which is also clearly incorrect. You should replace the color scale of these figures with the correct scale.

2. The information indicated in subsection 4.3.2, the time complexity, is insufficient. Do these times correspond to the training time? To the test time? What is the average time per image? What programming language/framework/libraries were used? Did you not use a GPU? In the caption, you should say "seconds" instead of "Second". And you should say "Indian Pines dataset" and "Pavia University dataset".

Author Response

The authors would like to thank the reviewer’s for your constructive comments and suggestions to improve the quality of the manuscript. Your suggestions helped us in improving the content of the paper significantly. We have studied all the comments  carefully and made revisions, which is highlighted with yellow color in the revised version. A point-wise response to the comments follows.
